# MULTI-AGENT GENERATIVE ADVERSARIAL IMITATION LEARNING

**Jiaming Song, Hongyu Ren, Dorsa Sadigh & Stefano Ermon**
Computer Science Department
Stanford University

## ABSTRACT

We propose a new framework for multi-agent imitation learning for general Markov games, where we build upon a generalized notion of inverse reinforcement learning. We introduce a practical multi-agent actor-critic algorithm with good empirical performance. Our method can be used to imitate complex behaviors in high-dimensional environments with multiple cooperative or competitive agents.

## 1 MARKOV GAMES AND IMITATION LEARNING

We consider an extension of Markov decision process (MDPs) called Markov (stochastic) games (Littman, 1994; Lowe et al., 2017). A Markov game (MG) for $N$ agents is defined by $N$ sets of states $\{\mathcal{S}_i\}_{i=1}^N$ and $N$ sets of actions $\{\mathcal{A}_i\}_{i=1}^N$ for each agent respectively. We let $S = \prod_i S_i$ represent the set of states. The function $T : \mathcal{S} \times \mathcal{A}_1 \times \cdots \times \mathcal{A}_N \to \mathcal{P}(\mathcal{S})$ describes the transition between states, where $\mathcal{P}(\mathcal{S})$ denotes the set of probability distributions over $\mathcal{S}$. Each agent $i$ receives a reward given by a function $r_i : \mathcal{S} \times \mathcal{A}_1 \times \cdots \times \mathcal{A}_N \to \mathbb{R}$, and aims to maximize its own total expected return $R_i = \sum_{t=0}^H \gamma^t r_{i,t}$, where $\gamma$ is the discount factor and $H$ is the time horizon, by selecting actions through a stochastic policy $\pi_i : \mathcal{S}_i \times \mathcal{A}_i \to [0, 1]$.

Our goal in imitation learning is to learn policies that behave similar to collected expert demonstrations under the assumption that we don't have access to the reward values. The expert demonstrations form a dataset of state-action pairs $\mathcal{D} = \{(s_j, a_j)\}_{j=1}^M$, which are collected by sampling $s_0 \sim \eta(s), a_t = \pi_E(a_t|s_t), s_{t+1} \sim T(s_{t+1}|s_t, a_t)$. We assume all experts operate in the same environment, and that once we obtain demonstrations $\mathcal{D}$, we cannot ask for further expert interactions with the environment (unlike in (Ross et al., 2011) or (Hadfield-Menell et al., 2016)).

## 2 MULTI-AGENT GENERATIVE ADVERSARIAL IMITATION LEARNING

In this work, we consider a distributed setting for imitation learning, where we leverage ideas from (Ho & Ermon, 2016) for a novel algorithm for Multi-Agent Generative Adversarial Imitation Learning (MAGAIL). For each agent $i$, we have a discriminator (denoted as $D_{\omega_i}$) mapping state action-pairs to *scores*. The discriminators are optimized to discriminate expert demonstrations from behaviors produced by $\pi_i$. Implicitly, $D_{\omega_i}$ plays the role of a reward function for the generator, which in turn attempts to train the agent to maximize its reward thus fooling the discriminator. Hence we optimize the following objective for each agent $i$:

$$\min_{\theta_i} \max_{\omega_i} \mathbb{E}_{\pi_{\theta_i}}[\log D_{\omega_i}(s, a_i)] + \mathbb{E}_{\pi_{E_i}}[\log(1 - D_{\omega_i}(s, a_i))]$$

This process is similar to coordinate descent in the sense that each agent updates their policies independently. We update $\pi_\theta$ through multi-agent reinforcement learning. It is possible to encode our inductive bias about the reward structure through the discriminator learning process. We consider three types of priors for common multi-agent settings.

**Fully Cooperative.** The easiest case is to assume that the agents are fully cooperative, i.e. they share the same reward function. One could argue this corresponds to the GAIL case, where the RL procedure is operated on multiple agents, so conclusions in (Ho & Ermon, 2016) would still hold.

**Decentralized Rationality.** We make no assumptions over the correlation between the rewards, yet we assume experts are acting rationally under a reward function that depends only on their own observations and actions. This setup corresponds to having one discriminator for each agent which discriminates the trajectories as observed by agent $i$.

**Zero Sum.** Assume there are two agents that receive opposite rewards, so $r_1 = -r_2$. An adversarial training procedure can be designed using the following fact:

$$V(\pi_{E_1}, \pi_2) \geq V(\pi_{E_1}, \pi_{E_2}) \geq V(\pi_1, \pi_{E_2})$$

where $V(\pi_1, \pi_2) = \mathbb{E}_{\pi_1, \pi_2}[r_1(s, a)]$ is the expected outcome for agent 1 when the agents choose policies $\pi_1$ and $\pi_2$ respectively. The discriminator could maximize the reward for trajectories in $(\pi_{E_1}, \pi_2)$ and minimize the reward for trajectories in $(\pi_2, \pi_{E_1})$.

## 3 MULTI-AGENT ACTOR-CRITIC WITH KRONECKER FACTORS

Once we obtain rewards from the discriminator, we wish to use an algorithm for multi-agent RL that has good sample efficiency in practice, which then can result in effective updates at each step. We design our MARL algorithm based on Actor-Critic with Kronecker-factored Trust Region (ACKTR, Wu et al. (2017)), a state-of-the-art actor-critic method in deep RL.

Our algorithm, which we refer to as Multi-agent Actor-Critic with Kronecker-factors (MACK), uses the framework of centralized training with decentralized execution (Foerster et al., 2016; Lowe et al., 2017); policies are trained with additional information to reduce variance (only in training). We let the advantage function of every agent be a function of all agents' observations and actions:

$$A_{\phi_i}^{\pi_i}(s_t, a_t) = \sum_{j=0}^{k-1} \left( \gamma^j r(s_{t+j}, a_{t+j}) + \gamma^k V_{\phi_i}^{\pi_i}(s_{t+k}, a_{-i,t}) \right) - V_{\phi_i}^{\pi_i}(s_t, a_{-i,t})$$

where $V_{\phi_i}^{\pi_i}(o_k, a_{-i})$ is an estimated value function for $i$, when other agents take actions $a_{-i}$. This value function treats other agents as a part of the environment, whereas $(o_{-i}, a_{-i})$ serves as additional information for variance reduction. The policy gradient for agent $i$ is written as:

$$\nabla_\theta R_i(\theta_i) = \mathbb{E}_\pi \Big[ \sum_{t=0}^{T} \nabla_\theta \pi_{\theta_i}(a_{i,t}|s_{i,t}) A_{\theta_i}^{\pi_i}(s_t, a_t) \Big]$$

We use K-FAC to update both $\theta$ and $\phi$, but do not use trust regions to schedule the learning rate, since we find a linearly decaying learning rate schedule to have similar performance.

## 4 EXPERIMENTS

We consider the two-dimensional particle environment proposed in (Lowe et al., 2017), which consists of $N$ agents and $L$ landmarks. Agents may take *physical actions* and *communication actions* that get broadcasted to other agents. We consider two cooperative settings (all agents attempt to maximize a shared reward) and two that are competitive (agents have conflicting goals). All these environments have an underlying true reward that allows us to estimate the performance of our agents.

**Cooperative Communication:** two agents must cooperate to reach one of three colored landmarks. We consider an asymmetric observation setting: One agent ("speaker") knows the goal but cannot move, so it must convey the message to the other agent ("listener") that can move but does not observe the goal.

**Cooperative Navigation:** three agents must cooperate through physical actions to reach three landmarks; ideally, each agent should cover a single landmark.

**Keep-Away:** two agents have contradictory goals, where agent 1 tries to reach one of the two targeted landmarks, while agent 2 (the adversary) tries to keep agent 1 from reaching its target. The adversary does not observe the target, so it can only act based on agent 1's actions.

**Predator-Prey:** three slower cooperating adversaries must chase the faster agent; the adversaries are rewarded by touching the faster agent while that agent is penalized.

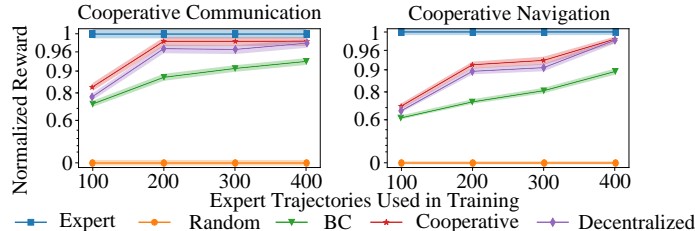

Figure 1: True reward obtained from cooperative tasks. Performance of experts and random policies are normalized to one and zero respectively. We use inverse log scale for better comparison.

For the cooperative tasks, we use an analytic expression defining the expert policy. For the competitive tasks, we use MACK to train expert policies based on the true underlying rewards. We then use the expert policies to simulate trajectories $\mathcal{D}$, which can then be used as demonstrations for imitation learning, where we assume both the underlying rewards and the expert policy are unknown. Following (Li et al., 2017), we pretrain MAGAIL using behavior cloning as initialization to reduce sample complexity.

## 4.1 COOPERATIVE TASKS

We evaluate performance in cooperative tasks via the average expected reward obtained by all the agents in an episode. In this environment, the starting state is randomly initialized, so generalization is crucial. We consider 100 to 400 episodes as expert demonstration, and display the performance of cooperative MAGAIL, decentralized MAGAIL and behavior cloning (BC) in Figure 1.

Naturally, the performance of BC and MAGAIL increases with more expert demonstrations. MAGAIL performs consistently better than BC in all the settings. Moreover, the cooperative MAGAIL performs slightly better than decentralized MAGAIL due to the better prior, but decentralized MAGAIL still manages to achieve reasonable performance.

### 4.1.1 COMPETITIVE TASKS

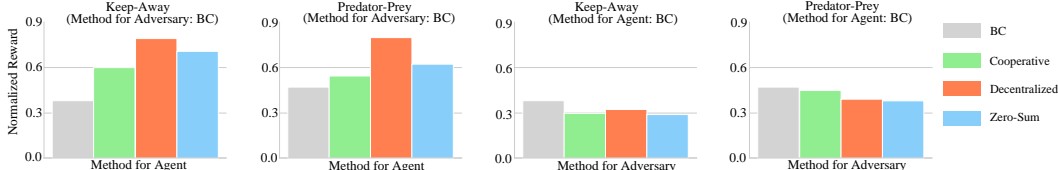

Figure 2: Normalized agent performance in competitive tasks. If the adversary is BC (charts 1 and 2), higher performance is better; if agent is BC, lower performance is better (charts 3 and 4).

We consider all three types of Multi-Agent GAIL (cooperative, decentralized, zero-sum) and BC in both competitive tasks. Since there are two opposing sides, it is hard to measure performance directly. Therefore, we compare the rewards by letting (agents trained by) BC play against (adversaries trained by) other methods, and vice versa. From Figure 2, decentralized and zero-sum MAGAIL often perform better than centralized MAGAIL and BC, which suggests that the selection of the suitable prior is important for good empirical performance.

## 5 CONCLUSION

We propose a model-free imitation learning algorithm for multi-agent settings that leverages recent advances in adversarial training; the algorithm is agnostic to the competitive/noncompetitive structure of the environment, but allows for incorporating a prior when it is available. Experimental results demonstrate that it is able to imitate complex behaviors in high-dimensional environments with both cooperative and adversarial interactions.

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
