# OpenReview forum: "Multi-Agent Generative Adversarial Imitation Learning"
_ICLR.cc/2018/Workshop — Accept_

### Official Review · AnonReviewer3 · 2018-03-11
**Limited to obvious cases**

**Rating:** 6
**Confidence:** 4

**Review:**

The authors propose to apply generative adversarial imitation learning to multi player Markov games.

My opinion is that the studied cases are the most simple cases in multi-agent problems. Rewards are either the same for every agents (collaborative) or opposite (zero-sum) or agents are totally independent. Therefore there is little challenge for imitation learning as in the two first cases, there is only one reward function to learn and in the third, each agent can be imitated independently from the others.

As the authors say, the computation of the advantage function assumes that other agents are just part of the environment which makes the problem much simpler.

---

> ### Public Comment · ~Jiaming_Song1 · 2018-03-21
> **Some Clarifications**
>
> Thank you for your comment! To clarify some of your concerns:
>
> - One challenge to inverse reinforcement learning is the ill-defined nature of the problem. Even in single agent settings there are multiple solutions to the IRL problem (for example, set zero reward everywhere). This is even worse in the multi-agent case because there could be several Nash equilibria. If we do not make any assumptions to simplify the problem, it is highly possible to learn many rewards that explain the demonstrated behavior. If we merely learn a new set of policies from any set of rewards, we may land into another Nash equilibrium that do not reflect the expert policies.

---

### Official Review · AnonReviewer1 · 2018-03-16
**Good paper overall, but seems like an incremental extension of the GAIL algorithm (Ho & Ermon, 2016)**

**Rating:** 6
**Confidence:** 4

**Review:**

The paper considers the problem of learning from demonstrations to act in a multi-agent Markov game. The proposed method is based on the GAIL algorithm, where a discriminator is trained to distinguish between expert trajectories and learned trajectories, which are generated by a generator network that tries to fool the discriminator. The proposed algorithm, named MAGAIL, extends GAIL to multi-agent systems. It seems like the extension consists in training all the agents centrally, and executing the learned policies in a decentralized fashion. Each agent considers the other agents as part of the environment while training. An advantage function is considered to tackle the high variance resulting from the non-stationary nature of the environment. Experiments are performed on several tasks, with varying levels of competition vs cooperation. The results indicate that the MAGAIL achieves a better performance than a simple Behavioral Cloning (BC).
I am concerned about the technical contribution of this work. The proposed algorithms seems like a straightforward extension of the GAIL. The authors should study the implications of considering other agents as part of the environment, which could hinder the learning process. It is also strange to consider such an assumption in the zero-sum case.

---

> ### Public Comment · ~Jiaming_Song1 · 2018-03-21
> **Some Clarifications**
>
> Thank you for your comment! We thought about most of your concerns as well, but chose to omit the details due to limited space. To clarify:
>
> - One interesting notion in the multi-agent case is that there might not be optimal policy, and we need to replace this notion with equilibrium concepts such as Nash equilibrium. For certain reward structures, multiple NE would exist; this would require us to rethink the notion of inverse reinforcement learning in the multi-agent settings. We did not mention this line of reasoning (and how we obtain MAGAIL from here) due to space constraints.
>
> - Our proposed multi-agent actor critic with K-FAC did not consider other agents as merely part of the environment. The critic considered the observations and actions of other agents, similar to MADDPG. We find that merely providing such external data is able to train efficiently as opposed to using/inferring other agent's policies as in MADDPG.

---

### Decision · Program_Chairs · 2018-03-20
**ICLR 2018 Workshop Acceptance Decision**

**Decision:**

Accept

**Comment:**

Congratulations, your paper was accepted to the ICLR workshop.